# Supportive, Delegated, and Common Dyadic Coping Mediates the Association between Adult Attachment Representation and Relationship Satisfaction: A Dyadic Approach

**DOI:** 10.3390/ijerph19138026

**Published:** 2022-06-30

**Authors:** Anna Wendołowska, Małgorzata Steć, Dorota Czyżowska

**Affiliations:** 1Institute of Psychology, Jagiellonian University, 30-060 Kraków, Poland; d.czyzowska@uj.edu.pl; 2Institute of Psychology, Jesuit University Ignatianum, 26 Kopernika Street, 31-501 Kraków, Poland; malgorzata.stec@ignatianum.edu.pl

**Keywords:** attachment, relationship satisfaction, dyadic coping, actor–partner interdependence mediation model, adult attachment interview

## Abstract

The aim of this study was to examine intrapersonal (actor) and interpersonal (partner) associations between attachment, assessed by the Adult Attachment Interview, and satisfaction with the relationship, as well as to establish the possibility of the mediatory effect of supportive, delegated, and common dyadic coping on the aforementioned associations. A dyadic approach has been introduced, using the actor–partner interdependence mediation model and data from 114 heterosexual couples, aged 26 to 60. It has been shown that one’s own secure attachment can be perceived as the predictor of one’s own relationship satisfaction in women and men and the predictor of a partner’s relationship satisfaction in men. The findings support the partially mediating role of dyadic coping in the association between attachment and relationship satisfaction and are a significant contribution to the issue of dyadic coping in general. Adults’ secure representations of their childhood experiences may be effective in using their partners as a secure base and also in serving as a secure base themselves, but it is not the sole influence on the quality of the couple’s experience together. The we-ness phenomenon and resulting clinical implications were discussed.

## 1. Introduction

Romantic relationships are significant for personal and family well-being [1,2,3]. Low marital quality is associated with negative stress-related outcomes, including high blood pressure [4], poor immune system [5], mortality [6], and mental health problems [7,8], whereas a good relationship quality plays the role of a protective factor [9].

Romantic relationship quality can be reflected by various indicators, such as relationship satisfaction [10,11], commitment [12,13], emotional experience [14], conflict resolution [15], love and closeness [16], tenderness, communication [17], and so on. According to Plopa and Rostowski [18], harmonious development of marriage is determined by high intimacy between the spouses, their self-realization through marital roles and tasks, similarity in fulfilling life goals, and low disappointment. The level of satisfaction with the relationship is the result of effective communication, negotiation, and compromise on the principles of coexistence: needs, goals, values, and lifestyle.

To support healthy relationships, it is important to understand the predictors of relationship quality for individuals and their partners. Attachment is repeatedly identified as key predictor of relationship quality [19,20,21,22], relationship satisfaction [23,24], commitment [25,26], intimacy [27,28], sexual satisfaction [29,30], marital adjustment [31], and relationship stability [32,33]. There is also a link between attachment insecurities and poor coping skills [34,35], more perceived conflict [36], negative affect [37], pessimistic attribution [38], and relationship distress [39].

Some findings provide evidence that both one’s own and partner’s attachment patterns are relevant to relationship satisfaction [40]; however, studies report inconsistent results. Hamidi [41] found a positive relationship between secure attachment and marital satisfaction, Kamel Abbasi et al. [42] found that marital satisfaction was negatively linked with insecure attachment and positively linked with secure attachment, but Mohammadi et al. [43] found no relationship between secure attachment style and marital satisfaction. While some results indicated that the link between insecure attachment and relationship satisfaction was not moderated by gender [44], there is also evidence that it may be gender-specific [45]. When controlling for attachment, females presented lower levels of marital satisfaction then males [22]. Relationship satisfaction was positively associated with secure attachment and negatively with avoidant attachment for males and females, but ambivalent attachment was linked to low satisfaction only for females [46]. Preoccupied attachment had stronger negative effects on perceived relationship quality for females than other attachment patterns [47]. In other studies, men who are high in avoidance and men and women who are high in anxiety reported lower relationship quality [48], but secure attachment was only positively linked with male satisfaction [45,49]. Additionally, the perception of partner’s avoidance was associated with relationship satisfaction more for females than for males [25].

Four main attachment categories were identified: secure, insecure ambivalent/anxious, insecure avoidant, and disorganized [50]. Sensitivity, responsiveness, and availability of parents promote attachment security, which is the basis of emotional self-control and a protective factor in stressful situations [51]. Repeated infant experiences of parental rejection and neglect foster avoidant attachment, characterized by discomfort with intimacy, mistrust of others, and suppressed emotional expression [52]. Incoherent or inconsistent care from attachment figures fosters ambivalent attachment, characterized by chronic self-doubt and fear of abandonment [53]. Infant’s perceptions of their caregivers as “frightened or frightening,” lead to the fourth attachment category, disorganization [54], which is to protect against difficult content that is pushed into the unconscious. It exhibits conflicted behaviors that indicate inability to maintain one coherent attachment strategy in the face of distress [55]. Attachment theory was originally designed to explain the emotional bond between infants and their primary caregivers; however, attachment representations, through the internalizing of the attachment figure [56], exert their influence beyond childhood experiences and are strongly predictive of other close relationships later in life [53,57,58,59]. Early attachment relationships are the basis of emotional regulation skills and provide a template to form subsequent relationships in model-consistent and model-confirming ways, resulting in relatively stable attachment orientations over time [53,56,60,61]. Interpersonal representations of secure attachment are internalized by those raised in secure family environments, which favor the development of responsive and supportive attachment figure qualities that can be activated in future healthy adult relationships.

### Dyadic Coping as a Mediator between Attachment and Relationship Satisfaction

Dyadic coping (DC) is a transactional process that involves both members of the dyad. The partners send verbal or non-verbal signals to communicate stress to one another, and induce response to cope together [62,63]. Positive dyadic coping refers to one partner’s support-seeking followed by another partner’s emotional and instrumental help (supportive DC) or taking over certain tasks and duties in order to reduce his or her stress (delegated DC). Positive dyadic coping may also involve shared problem-solving and mutual consolation (common DC). Negative dyadic coping is manifested by superficial, ambivalent, or hostile behaviors in response to the partner’s stress [64].

DC is closely linked to couples’ relationship functioning [64,65,66]. Coping in a relationship is an interdependent process. Relationship satisfaction, quality, and stability largely depend on whether both partners are able and willing to provide effective and sensitive care to one another [67]. In the romantic context, partners constantly alternate from support seekers to support providers, which is particularly difficult for people with an insecure attachment [68]. Avoidant individuals disengage their attempts to seek proximity and suppress their need for support; they tend to avoid conflicts by distancing and becoming task-focused [22]. Ambivalent individuals, in fear of abandonment, are overly focused on relationship and the partner, often provide ineffective help, and do not appreciate the received support [67,69].

Positive DC was related to lower stress, better relationship functioning, and higher couple satisfaction in the face of daily hassles [63], traumatic experiences [70], and chronic stress [71]. Negative DC was associated with detrimental outcomes [72,73]. Associations between dyadic coping and relationship quality were also true for cross-partner perceptions of relationship quality [71,74].

## 2. Aim of the Study

Our study makes several contributions to the literature. Research indicates that DC mediates the individual coping efforts and is a stronger predictor of relationship satisfaction than individual coping [75]; it also mediates the relationship between emotional intelligence and marital quality [76]. However, assessment of the mediating role of DC in the link between adult attachment and relationship satisfaction has yielded inconsistent findings. To this date, one other study has examined the role of DC as a possible mediator of the relationship between attachment and relationship satisfaction [77], but did not find any significant mediation effects. General DC partially mediated the association between insecure attachment and marital satisfaction, but the study sample was limited to a student population and only used data from one partner [78]; common DC was a mediator between insecure attachment and parental adjustment to the first year postpartum [79], and also between insecure attachment and quality of life in an ovarian cancer patient population [80]. Our findings expand this line of research by (1) assessing current states of mind with respect to attachment versus self-reported attachment measures, (2) using a community sample versus student samples, (3) implementing dyadic data instead of individual data, and (4) testing the separately mediating role of positive DC forms instead of total DC or total positive DC.

Our first goal was to examine whether a person’s secure attachment predicts his or her own relationship satisfaction (actor effect) and his or her partner’s relationship satisfaction (partner effect). Based on the results of the research to date [16,20,81], we expect significant positive effects of one’s own secure attachment on one’s own relationship satisfaction (Hypothesis 1a (H1a)) and significant positive effects of one’s own secure attachment on a partner’s relationship satisfaction (Hypothesis 1b (H1b)).

Our second goal was to test whether selected forms of positive dyadic coping—supportive DC, delegating DC, and common DC—mediate the actor and partner effects of attachment on relationship satisfaction. We tested both forms of supportive and delegated DC: self-perception and other-perception. Given the evidence that attachment security is associated with a higher dyadic coping [78,82] and that dyadic coping influences relationship satisfaction [73,83], it was hypothesized that positive dyadic coping (self- and other-perceived delegated DC, self- and other-perceived supportive DC, and common DC) as mediator would better explain the process through which secure attachment and relationship satisfaction are related (Hypothesis 2a–e (H2a–e)). Experience of availability and responsiveness from primary attachment figures during childhood may influence the ability to accept and provide support within close adult relationships [84].

## 3. Materials and Methods

### 3.1. Participants

A total of 114 Polish heterosexual couples participated in the study (N = 228 individuals). On average, men were 32.83 years old (SD = 10.29) and women were 31.52 years old (SD = 10.20). In terms of education, 45.2% of men and 46% of women had completed secondary school education, whereas 35.7% of men and 53.7% of women held a high school degree. Most of the men (85.2%) were employed, 7.8% were students, 3.5% were unemployed and 1.4% were retired; 69.6% of women were employed, 22.6% were university students, and 3.5% were unemployed. All couples had been together in a committed relationship for a least a year, and shared a household; 59% had no children, 13.9% had 1 child, and 23.3% had 2 and more children. Most of the couples (44.3%) had been in their current relationship for up to 5 years, 21.7% for up to 10 years, and 12.2% for more than 30 years. 34% of the couples were in an informal relationship and 63.4% were married. Only 7.1% of men and 8.8% of women rated their financial situation as bad; 65.2% of men and 60.6% of women rated it as good or very good.

### 3.2. Procedures

Participants were recruited in various places using traditional methods such as leaflets, advertisements, and email. To participate in the study, both partners had to be at least 25 years old and in their current committed relationship for at least 1 year. Informed consent was obtained from all individual participants in the study. Partners were interviewed separately by an interviewer trained to conduct the AAI. The interviews were audio-recorded. After the interview, each partner completed a set of self-report questionnaires. All participants were remunerated with cinema tickets for their time and effort. The transcripts were subsequently coded by a certified coder, trained in the AAI coding system [82]. All procedures performed in studies involving human participants were in accordance with the ethical standards of the institutional research committee (the Ethics Committee at the Institute of Psychology, Jagiellonian University; KE/01/102018) and the 1964 Helsinki declaration and its later amendments or comparable ethical standards.

### 3.3. Measures

Relationship satisfaction. The Matched Marriage Questionnaire (KDM-2) [83] examines the level of relationship satisfaction (in the perception of both partners) in four dimensions: intimacy (e.g., “I like the nature of my partner”), self-realization (e.g., “On a basis of my marital experiences I think that we can find the fulfilment of our life only in marriage”), similarity (e.g., “Acting together and achieving common plans bring me satisfaction”, disappointment (e.g., “I regret my independence and freedom from the premarital stage”. The questionnaire consists of 32 items rated on a 5-point Likert scale from 1 (strongly agree) to 5 (strongly disagree). The total scores of the subscales give the overall level of satisfaction with the relationship. The tool has satisfactory psychometric properties. Its internal consistency is Cronbach’s α = 0.89.

Dyadic Coping. Dyadic Coping Inventory (DCI) [54] in Polish validation [84], is a tool with high reliability and psychometric validity; it examines communication between the partners and various forms of DC: supportive, delegative, negative, and common DC. The questionnaire consists of 37 statements which the subject rates on a 5-point scale from 1 (“very rarely”) to 5 (“very often”). In our study, self-perceived supportive DC (e.g., I show my partner compassion and understanding), other-perceived supportive DC (e.g., My partner helps me to see stressful situations in a different light), self-perceived delegated DC (e.g., I take on things that my partner would normally do in order to help him/her out), other-perceived delegated DC (e.g., When I am too busy, my partner helps me out), and common DC (e.g., Both of us try to face problems together and find specific solutions together) were used. The internal consistencies of the subscales are, respectively, α = 0.78, 0.83, 0.78, 0.84, 0.83.

Attachment. Attachment can be conceptualized categorically [65,66] or dimensionally [67], for both theoretical and methodological reasons [66,68]. We decided to use the Adult Attachment Interview (AAI) [69] in our study, as it is likely to capture individual’s potential capacities for caregiving, including highly specific aspects of emotional attunement and emotional regulation in dyads [70,71]. The AAI has been used in research on close relationships [65,72,73], and it continues to be the primary method of assessing adult attachment, despite the time-consuming training and coding of attachment interviews [70]. The AAI [82] is a well-known, semi-structured, one-hour interview for the assessment of current state of mind with respect to attachment. It probes adults about their early attachment experiences [85]. After transcription and coding of the AAI by a certified coder, according to the manual (Hesse, 2016), an attachment representation classification is given. The standard system for coding the AAI recommends classifying individuals categorically as having a secure (F), preoccupied (E), dismissing (Ds), or disorganized (U) state of mind. Those categories share conceptual overlap with secure, anxious-ambivalent, avoidant, and disorganized attachment in childhood, respectively. In organized attachment representations, there is one coherent mental strategy regarding attachment figures, either secure (F) or insecure (E, Ds). In disorganized (U) attachment representations, different, often contradicting, mental strategies are used.

Demographic questionnaire was used to collect demographic data, including age, gender, length of relationship, type of relationship (formal or informal), number of children, employment status, education level, and financial situation, which was rated from “poor” to “very good”.

### 3.4. Analysis Strategies

Individual differences in terms of attachment were assessed categorically, using the adult attachment interview (AAI). The intra-rater reliability was computed through Cohen’s kappa coefficient (k) [85] to verify the degree to which one rater’s coding of the same transcript remained constant at two different times. In this study, we realized the second coding for a sample of 50% of the transcripts after 2 years. The interpretation of kappa values was based on the Landis and Koch guidelines: values < 0 as indicating no agreement and 0–0.20 as slight, 0.21–0.40 as fair, 0.41–0.60 as moderate, 0.61–0.80 as substantial, and 0.81–1 as almost perfect agreement [86].

Preliminary analyses used IBM SPSS Statistics 26 statistical package (Armonk, NY, USA) delivered by Predictive Solutions (PS IMAGO PRO Academic package). The means with standard deviation were calculated for all continuous variables. The *t*-test for paired samples was used to analyze differences between men and women, and the *t*-test for independent samples was used to test differences between the groups of secure and insecure individuals. To avoid the problem of dependence between the partners’ observations, each analysis was performed separately for men and women. For dyadic and correlation analysis the AAI attachment categories were converted to a dichotomous variable, so that either a participant was insecure (0) or secure (1).

Taking into account the interdependence of the dyadic data, subsequent analyses were completed using the actor–partner interdependence model (APIM) [87] and the actor–partner interdependence mediation model (APIMeM) [88]. Using standard statistical methods with dyadic data negatively influences tests of significance because of biased standard errors and loss of degrees of freedom [89]. The APIM approach allows simultaneous estimation of effects for both members of the dyad while controlling for the interdependence between them [89]. In the basic APIM, two predictor variables (i.e., male’s attachment and female’s attachment) and two outcome variables (i.e., male’s relationship satisfaction and female’s relationship satisfaction) are included. The effect of each individual’s attachment on his or her own relationship satisfaction is represented by the actor effects; the effect of each individual’s attachment on his or her partner’s relationship satisfaction is represented by the partner effects [88]. The APIMeM is similar to APIM; however, apart from two predictors and two outcome variables, two mediators are included: one mediator for each member of a dyad (Figure 1).

In APIMeM, potentially four direct effects can be mediated [88]: (1) the actor effect of the male’s attachment on his satisfaction (a3^M^), (2) the actor effect of the female’s attachment on her satisfaction (a3^W^), (3) the partner effect of the male’s attachment on the female’s satisfaction (p3^W^), (4) the partner effect of the female’s attachment on the male’s satisfaction (p3^M^). Each of these direct effects possibly has two indirect effects: (1) an actor–actor indirect effect in which the actor effect is mediated by the actor mediator (e.g., the effect of male’s attachment on his relationship satisfaction mediated by his DC; a1^M^a2^M^); (2) a partner–partner effect in which the actor effect is mediated by the partner mediator (e.g., the effect of male’s attachment on his relationship satisfaction mediated by female’s DC; p1^W^p2^M^); (3) a partner–actor effect in which the partner effect is mediated by the actor mediator (e.g., the effect of male’s attachment on female’s relationship satisfaction mediated by male’s DC; a1^M^p2^W^); (4) an actor–partner effect in which the partner effect is mediated by the partner mediator (e.g., the effect of male’s attachment on female’s relationship satisfaction mediated by female’s DC; p1^W^a2^W^). Overall, there are eight possible indirect effects. We tested a model with the effects constrained to be equal across gender, which reduces the number of possible indirect effects from eight to four. If the test of distinguishability, including three equal means, and three equal variances is not statistically significant, dyad members can be treated as indistinguishable. The combined mediation test involved comparing a model with the four indirect effects to a model without the indirect effects. If the test is significant, it provides evidence of mediation. We examined whether positive forms of DC (self- and other-perceived supportive DC, self- and other-perceived delegated DC, and common DC) significantly mediated the association between attachment and relationship satisfaction. Separate analyses were completed for each of the selected positive forms of DC. A total of five APIMeMs were completed.

All APIM analyses were performed using lavaan, an R-package for structural equation modeling (SEM) [90]. The Monte Carlo method, or parametric bootstrap, with 40,000 trials was used for the indirect effects. To test for significance of the indirect effects, we estimated the bootstrap 95% confidence intervals. All tests were performed at the 0.05 significance level. The hypothesized models were evaluated using goodness-of-fit indices that included the chi-square, root mean square error of approximation (RMSEA; acceptable fit < 0.08), and standardized root mean squared residual (SRMR; acceptable fit < 0.08), supplementing with either Tucker-Lewis Index (TLI; acceptable fit > 0.95) or Comparative Fit Index (CFI; acceptable fit > 0.95) [91].

## 4. Results

We tested normality by examining skewness and kurtosis for all continuous variables (delegated DC self and other-perception, supportive DC self- and other-perception, common DC, and relationship satisfaction) separately for men and women. The result showed that the skewness is not larger than 0.89, and the kurtosis is not larger than 2.91. The value of skewness is smaller than 2 and the value of kurtosis is smaller than 7, suggesting that these data were approximately normally distributed [92].

Intra-rater reliability for the four-way AAI classification (F, E, DS, and U) was calculated for a sample of 50% of the transcripts and the result was 82.3%, showing an almost perfect agreement level (k > 0.81%) [86]. We examined the generalized attachment categories (AAI) of both sexes and our results did not reveal a significant difference between them in terms of four-way AAI classification (χ^2^ (3) = 2.14, *p* = 0.54). The distribution of AAI categories in our sample (women: 52.2% F, 26% Ds, 18.3% E, and 3.5% U; men: 53.9% F, 30% Ds, 13% E, and 2.6% U) was overlapping to those considered in other studies [93,94]. For further analyses, the AAI four categories were converted to a dichotomous secure–insecure variable.

Table 1 shows the descriptive statistics for tested continuous variables and gender differences. There were no significant differences between men and women in terms of supportive, delegated, and common DC and relationships satisfaction.

*T*-test was used to compare dichotomous attachment groups (secure and insecure), separately for women and men, for relationship satisfaction, and positive forms of DC (delegated DC, supportive DC, and common DC) (Table 2). Secure individuals of both sexes had significantly higher scores of relationship satisfaction and all forms of positive DC compared to insecure partners.

Both in the case of men and women, there were few moderate and strong correlations between the studied variables (Table 3). The strongest correlations in women we observe between secure attachment and relationship satisfaction, common DC, and self- and other-perceived delegated DC. Secure attachment in men is associated with their relationship satisfaction, common DC, and self-perceived delegated DC.

### 4.1. Secure Attachment as a Predictor of Relationship Satisfaction

We continued using attachment as a secure–insecure dichotomous variable for further dyadic APIM and APIMeM analysis. The minimum sample size to detect the actor and partner effects for an APIM analysis given a desired level of power of 0.80 and alpha of 0.05 is 81 dyads [95], making our sample of 114 dyads acceptable for the APIM analysis. At first, we examined actor and partner effects of secure attachment on relationship satisfaction using an actor–partner interdependence model [96]. The actor effect for women and men, and the partner effect for men, were found to be statistically significant (Figure 2). Secure attachment predicts higher relationship satisfaction in men and women (confirmed H1a). Female security of attachment predicts higher relationship satisfaction in their male partners (partially confirmed H1b). The basic actor–partner interdependence model is a saturated model (i.e., df = 0). We tested also whether participants’ age, financial status, and length of the relationship moderated the actor and partner effects. None of the tested covariates appeared to be significant for the attachment effects on relationship satisfaction.

#### Dyadic Coping as Mediator of a Relationship between Secure Attachment and Relationship Satisfaction

We tested separately five models with different mediators: self-perceived supportive DC, other-perceived supportive DC, self-perceived delegated DC, other-perceived delegated DC, and common DC. Thus, attachment was considered predictor, relationship satisfaction as outcome, and dyadic five different forms of DC were used as mediators.

First, we tested a model including self-perceived supportive DC as mediator (Table 4). As the test of distinguishability was not statistically significant (χ^2^ (12) = 10.57, *p* = 0.566, RMSEA = 0.000), the dyad members were treated as if they were indistinguishable, and we examined the estimates of the model with cross-gender equality constraints. The combined mediation test was significant (χ^2^ (2) = 26.82, *p* < 0.001, RMSEA = 0.329), providing evidence of mediation. Two of four indirect effects were significant, with the exception of the indirect partner–partner effect and indirect actor–partner effect. One’s own attachment predicted own relationship satisfaction through own self-perceived supportive DC. Partner’s attachment predicted own relationship satisfaction through own self-perceived supportive DC. Moreover, when controlling for the indirect effects, the direct partner effect became nonsignificant (Hypothesis 2a). We compared the size of the indirect actor and partner effects with the size of the corresponding total effects. Regarding the actor effect, the overall indirect effect (0.11) accounted for 37% of the total effect (0.30). Regarding the partner effect, the overall indirect effect (0.19) accounted for 64% of the total effect (0.17). The fit of the constrained model was very good (χ^2^ (4) = 5.40, TLI = 0.96, the CFI = 0.98, SRMR = 0.039, RMSEA = 0.055).

Second, we tested whether other-perceived supportive DC mediated the effects of attachment on relationship satisfaction (Table 5). As the test of distinguishability was not statistically significant (χ^2^ (12) = 7.15, *p* = 0.848, RMSEA = 0.000), we examined the model that included cross-gender equality constraints. The combined mediation test was significant (χ2 (2) = 37.25, *p* < 0.001, RMSEA = 0.391), providing evidence of mediation. Four out of four indirect effects were significant. Both, own, and the partner’s attachment predicted own and the partner’s relationship satisfaction through own and the partner’s other-perceived supportive DC. When controlling for the indirect effects, the direct partner effect became nonsignificant (Hypothesis 2b). We compared the size of the indirect actor and partner effects with the size of the corresponding total effects. Regarding the actor effect, the overall indirect effect (0.13) accounted for 31% of the total effect (0.43). With regard to the partner effect, the overall indirect effect (0.23) accounted for 79% of the total effect (0.17). The constrained model achieved a very good fit (χ^2^ (4)= 6.48, TLI = 0.95, CFI = 0.98, SRMR = 0.049, RMSEA = 0.074).

Third, we tested whether self-perceived delegated DC mediated the effects of attachment on relationship satisfaction (Table 6). The test of distinguishability was not statistically significant (χ^2^ (12) = 5.30, *p* = 0.947, RMSEA = 0.000), so the model that included cross-gender equality constraints was examined. The combined mediation test was significant (χ^2^ (3) = 48.20, *p* < 0.001, RMSEA = 0.362), providing evidence of mediation. Partner–partner and partner–actor indirect effect were insignificant. Own attachment predicted own relationship satisfaction through own self-perceived delegated DC. Partner’s attachment predicted own relationship satisfaction through the partner’s self-perceived delegated DC. When controlling for the indirect effects, the direct actor and partner effects became nonsignificant (Hypothesis 2c). Regarding the actor effect, the overall indirect effect (0.13) accounted for 30% of the total effect (0.43). Regarding the partner effects, the overall indirect effect (0.23) accounted for 74% of the total effect (0.12). The fit values of the constrained model for the TLI and RMSEA were slightly worse than the normative values specified by Hu and Bentler (1999): (χ^2^ (4)= 12.09, TLI = 0.80, CFI = 0.91, SRMR = 0.067, RMSEA = 0.133).

Next, we included other-perceived delegated DC as mediator (Table 7). We examined the model that included cross-gender equality constraints, as the test of distinguishability was not statistically significant (χ^2^ (12) = 10.78, *p* = 0.548, RMSEA = 0.000). The combined mediation test was significant (χ^2^ (3) = 17.53, *p* < 0.001, RMSEA = 0.260), providing evidence of mediation. Partner–partner and partner–actor indirect effect appeared to be insignificant. Own attachment predicted own relationship satisfaction through own other-perceived delegated DC. Partner’s attachment predicted own relationship satisfaction through the partner’s other-perceived delegated DC. When controlling for the indirect effects, the direct actor and partner effects became nonsignificant (Hypothesis 2d). Comparing the size of the indirect actor and partner effects with the size of the corresponding total effects, we observe that with regard to the actor effect, the overall indirect effect (0.05) accounted for 12% of the total effect (0.43). Regarding the partner effects, the overall indirect effect (0.09) accounted for 35% of the total effect (0.12). Again, the constrained model achieved a slightly worse fit (χ^2^ (4) = 12.99, TLI = 0.75, CFI = 0.99, SRMR = 0.069, RMSEA = 0.073).

The last tested model included common DC as mediator (Table 8). The test of distinguishability was not statistically significant (χ^2^ (12) = 14.01, *p* = 0.300, RMSEA = 0.038). We examined the model that included cross-gender equality constraints. The combined mediation test was significant (χ^2^ (2) = 34.50, *p* < 0.001, RMSEA = 0.376), providing evidence of mediation. Out of four possible indirect effects, only actor–actor indirect effect appeared significant. Own attachment predicted own (but not the partner) relationship satisfaction through common DC. When controlling for the indirect effects, the direct partner effect became nonsignificant (Hypothesis 2e). Comparing the size of the indirect actor and partner effects with the size of the corresponding total effects, we observe that with regard to the actor effect, the overall indirect effect (0.18) accounted for 43% of the total effect (0.43). Regarding the partner effects, although actor–partner and partner–actor indirect effects appeared to be insignificant, the total indirect effect (0.36) was significant and accounted for 53% of the total effect (0.12). The constrained model fit very well to the data (χ^2^ (4) = 4.59, TLI = 0.99, CFI = 0.99, SRMR = 0.044, RMSEA = 0.036).

## 5. Discussion

The aim of this study was to examine intrapersonal (actor) and interpersonal (partner) associations between attachment and relationship satisfaction and to explore whether these associations were mediated by self-perceived supportive DC, other-perceived supportive DC, self-perceived delegated DC, other-perceived delegated DC, and common DC. A dyadic approach to data analysis was implemented to simultaneously estimate actor and partner effects. We examined the generalized attachment categories (AAI) of both sexes and our results did not reveal a significant difference between them in terms of four-way AAI classification, which is in line with other research [94,97].

Secure individuals of both sexes had significantly higher levels of relationship satisfaction and higher scores in all forms of positive DC compared to insecure ones; secure attachment, relationship satisfaction, and dyadic coping correlated significantly. As hypothesized, own secure attachment predicted own relationship satisfaction in men and women (Hypothesis 1a), and it also predicted partner’s relationship satisfaction (Hypothesis 1b), but the latter was true for men only. This is partially in line with other studies that found actor and partner effects for attachment and relationship quality [20,35,81,98]. Secure individuals report higher relationship quality, as well as those whose with secure partners, both as assessed by the AAI [99] and by the self-report measures [16,20,81]. In a stable romantic relationship, the partners serve each other as secure base and play the role of a primary source of physical and emotional safety [100]. Secure partners tend to be available and responsive to fulfil each other’s attachment needs, enhancing satisfaction with the relationship [101]. Insecure attachment has been associated with lower trust and confidence [46], which may result in negative responses to the partner’s stress and can confirm their core concerns and thus negatively affect relationship satisfaction [69]. In our study however, the partner effect of male’s attachment on female’s relationship satisfaction was not significant. It was expected that an insecure partner would impact females’ satisfaction [20,36]; however, in most studies, the partner effect of attachment on relationship satisfaction was weaker compared to the actor level [35,102,103]. Further, in one study, having a high-anxiety or -avoidance husband did not have an impact on the marital adjustment of securely attached wives [11]. One possible explanation would be that as women more often find meaning in daily sacrifices and family work [104,105], they are focused on the durability of the relationship, care for relationship harmony [106], and are more able to forgive [107]. In other studies, women generally report higher life satisfaction than men [108]. There might be other factors for females that can foster their life satisfaction [16], like for example other-oriented attitude as opposed to more self-oriented men [109] or wider network of social support [110].

As hypothesized, our findings provide support for the partially mediating role of DC in the association between attachment and relationship satisfaction (Hypothesis 2a–e). In terms of the actor effects, secure attachment was positively associated with relationship satisfaction through a self- and other-perceived supportive DC, self- and other-perceived delegated DC, and common DC (actor–actor effects), and through other-perceived supportive DC by partner (partner–partner effect). Total indirect actor effect size of common DC was larger than the other tested positive DC behaviors, but still smaller than the direct effect. In all five tested models, direct actor effects remained significant, which confirms strong effect of attachment on relationship satisfaction. Regarding the partner effects, the partner’s secure attachment was associated with own higher relationship satisfaction through the partner’s self-perceived supportive DC, own self-perceived delegated DC, and by both: own and the partner’s other-perceived supportive DC. In all five tested models, partner indirect effects were greater than the corresponding direct effects. The total indirect partner effect size of other-perceived supportive DC (78% of the total effect), self-perceived delegated DC (74% of the total effect), and self-perceived supportive DC (64% of the total effect) was larger than the total indirect partner effect size of common DC (53% of the total effect) and other-perceived delegated DC (35% of the total effect). There is a close link between an individual’s attachment disposition and their ability to regulate their emotions [53] and cope during stressful events [111]. Secure individuals tend to cope with stress by engaging in problem-solving and by getting support from attachment figures [101]. Securely attached individuals’ tendency to be available and responsive may explain readiness to provide emotional and instrumental help to the partner, to engage in tasks that usually belong to the partner, to relieve him/her in face of stress, and also to recognize and appreciate the partner’s support.

It was interesting to test which form of the five tested positive DCs serve to amplify the relationship between attachment and relationship satisfaction the most. On the basis of previous research, common DC is the strongest predictor of relationship satisfaction [68,69,70]. Many studies also point to the role of a supportive DC, both self- and other-perceived, on relationship functioning [70,71,72]. Few studies have so far focused on delegated DC, which is taking over tasks upon request to relieve the partner’s stress [48]. The more delegated DC the partners provided to the patient coping with breast cancer, the fewer depressive symptoms they experienced. The more delegated DC patients provided to the partner, the more depressive symptoms they experienced. The partners of the patients experienced also more depressive symptoms the more supportive DC the patient provided to them [73]. In our study, self and other-supportive DC serves “best” to explain the process through which attachment and relationship satisfaction are related. Supportive DC refers to one partner’s attempts to help the other partner in their coping efforts by focusing on the problem (e.g., giving advice) or emotions (e.g., showing compassion). Attachment influences relationship satisfaction by promoting DC supportive behavior as components of a secure base [112].

Studies suggest that individuals who provide support to a stressed partner report increased well-being [113], relationship satisfaction [109,114,115,116,117], and stability [118]. DC also enhances relationship functioning [119], constructive conflict resolution [120], and buffers the negative effects of stress [121]. In terms of health conditions, use of supportive DC decreases distress [71,122] and depression symptoms [123]; one study found that in women with breast cancer, higher supportive DC offered by their partners increased their depressive symptoms [124]. Subjective perception of a supportive partner seems even more important form relationship satisfaction than providing support [125,126].

The important role of self-perceived delegated DC should also be emphasized. It is a specific form of support that differs from supportive DC. Individual not only offers a support to his/her partner, but also takes over tasks and responsibilities to relieve the partner’s stress upon request. In our study, the effect size of self-perceived delegated DC (0.23) was the same as of other-perceived supportive DC. It has been suggested to be rather small [127]; however, satisfaction with the relationship is a multidimensional concept that might be affected by multiple factors [1], so the predictive power of each single factor is necessarily limited. Delegated DC, compared to other DC dimensions, was less often the focus of researchers. Generally, delegated DC is positively linked to individual positive coping strategies [128,129], constructive conflict resolution, and relationship satisfaction [115,116,120,130] and negatively associated with couple communication [117] and satisfaction with the relationship [128]. In couples coping with illness, delegated DC provided by the patients to their partners might negatively impact the patients’ quality of life [131] and increase their depressive symptoms [124]. As research shows, the ability to replace a partner in his/her duties is a complex competence that requires tact, good communication, skills, and empathy. Delegated DC has been included only in the Systemic–Transactional Model studies [62], and its measurement instrument, the DCI [132]. Its role may be underestimated due to the communication component, being part of the stress communication scale in the DCI, and the fact that in several DCI validations [117,130,133], items from the stress communications scale referring to delegated DC (e.g., I ask my partner to do things for me when I have too much to do) have been deleted to achieve better fit. It would be worth taking a closer look at the delegated DC subscale and conducting further in-depth analyses.

This study presents some limitations. First, a national convenience sample was used, which could have resulted in a biased sample selection towards lengthier and happier relationships, which is true in probably most of the research on close relationships. Second, to our best knowledge, there is only one Polish-speaking AAI certified coder, and we necessarily had to rely on double-coding instead of two independent coders’ assessments. Third, other results were only based on self-report measures. Fourth, the correlational nature of this study precludes making inferences about causation in the observed associations. Fifth, a relatively small group of respondents does not allow for the analysis of more complex models and a broader generalization of conclusions. Finally, the assessment of dyadic coping was tied to general daily hassles, not any specific stressful event, and thus all generalizations must be treated with caution to avoid random possible relationships between variables. Behavioral observations, daily diaries, or physiological measurements with different samples, as well as longitudinal designs, could be used in future studies to further establish the link between attachment, dyadic coping, and relationship satisfaction in a romantic context.

## 6. Conclusions

The current study focused on the role of adult attachment representation and dyadic coping in shaping relationship satisfaction. Most of the recent studies have utilized the self-report measures [134], but it also appears that the generalized representations originating in childhood do play a role in influencing DC behavior in adult romantic relationships [112]. Many past studies focused on individual and intrapersonal analytic strategies. The dyadic approach, which was implemented in this study, considers the couple as the unit of analysis, which provides understanding of the dyadic complexity of the couple system. The APIM [135] allows consideration of the reciprocal influence of each partner on their own and their partner’s outcome simultaneously.

Attachment is a lifespan phenomenon, not only in the developmental context, but also in the context of close relationships, explaining the influence of attachment representations on secure base behavior towards a romantic partner [136,137]. Adult attachment representations, however, are not the sole influences on attachment behavior within romantic relationships. The current partner and romantic relationship experiences, particularly in time of stress or need, play a role in fostering the feeling of we-ness and intimacy [138]. Adults’ secure attachment representations of their childhood experiences may be effective in using their partners as a secure base and also in serving as a secure base themselves, which should result in a substantial difference in the quality of the couple’s experience together [112]. It seems that the main characteristics of adult attachment state of mind, such as integration of early experience, competence to evaluate and explain experience, and an overall valuing of attachment and coherency in the expression of needs and emotions can contribute to the development of higher dyadic coping competencies. Individuals with overly insecure or disorganized attachment patterns interact with others in overly distancing or demanding ways [139,140], which may increase risk of interpersonal conflicts, reduce the availability of support [141,142], and experience close relationships as stressful and unsatisfying [143].

The findings bear theoretical implications informing clinical interventions. Attachment strategies provide understanding whether and how distressed people express their need for support from their partners [21]. In conflict, insecure people tend to blame their partners for their low satisfaction [102,144]. In this study, the partner association between attachment and satisfaction was weaker than the actor association and became insignificant when controlling for dyadic coping. Emotionally Focused Couple Therapy aims at replacing insecure attachment strategies with the secure primary attachment strategy [145]. Enhancement of attachment security of the partners may bring benefits in terms of better emotional co-regulation, proximity-seeking behaviors, intimacy between partners, and more effective coping strategies [53]. The Couples Coping Enhancement Training [146] helps couples improve their emotional communication about daily hassles and implement dyadic coping strategies to change their negative responses for a positive transformation of self- and the other-perception.

## Figures and Tables

**Figure 1 ijerph-19-08026-f001:**
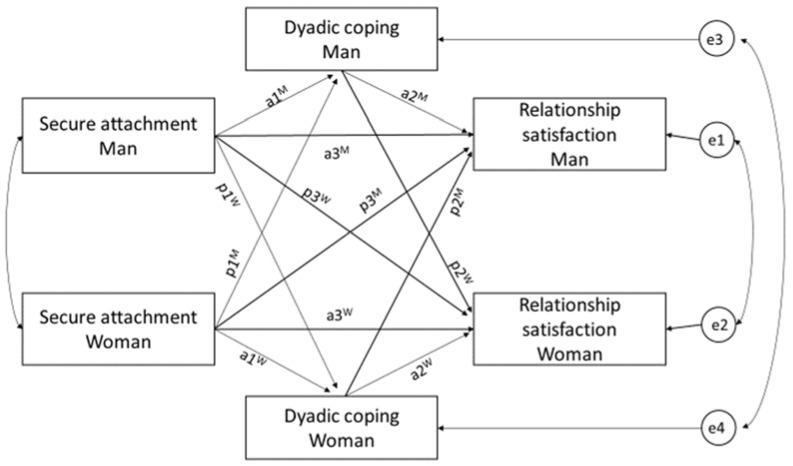
Actor–partner interdependence mediation model. Rectangles = predictor and outcome variables; circles = error terms (e1–4); curved double-headed arrows = covariances; a = actor effect; p = partner effect; W = women; M = men. Models were computed separately for five dyadic coping (DC) scales (self-perceived supportive DC and other-perceived supportive DC, self-perceived delegated DC, other-perceived delegated DC, and common DC) as mediators.

**Figure 2 ijerph-19-08026-f002:**
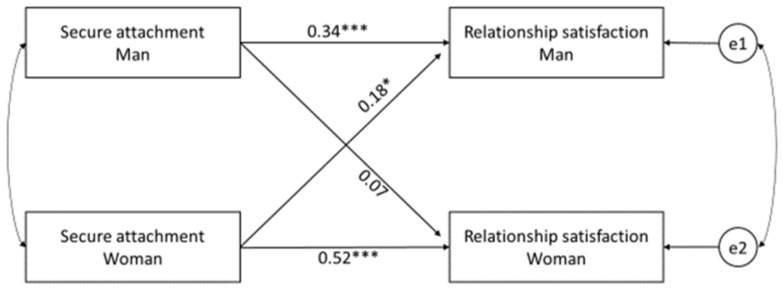
Secure attachment and relationship satisfaction (Model 1). Rectangles = independent and dependent variables; two circles = latent error terms (e1, e2: residual errors on satisfaction for males and females, respectively); the arrows = the actor and partner effects. The curved double-headed arrows on the left = covariances between the independent variables; the curved double-headed arrow on the right = correlation between the two error terms; * *p* < 0.05; *** *p* < 0.001.

**Table 1 ijerph-19-08026-t001:** Descriptive statistics and gender differences.

	M	SD	*t*-Test	*p*
Supportive DC self	15.92	2.29	1.91	0.52
	15.39	2.49		
Delegated DC self	4.10	0.84	0.29	0.77
	4.07	0.83		
Supportive DC other	19.18	3.88	−0.20	0.84
	19.26	3.89		
Delegated DC other	6.54	2.04	0.28	0.78
	6.47	1.78		
Common DC	19.40	3.83	0.48	0.63
	19.22	3.73		
Relationship satisfaction	125.76	15.19	−0.16	0.87
	126.01	14.72		

Women’s scores are in the upper row.

**Table 2 ijerph-19-08026-t002:** Differences between secure and insecure individuals.

	Women		Men	
	M	*t*-Test	M	*t*-Test
Supportive DC self	15.19	−3.41 ***	14.58	−3.35 ***
	16.58		16.08	
Delegated DC self	3.70	−5.26 ***	3.65	−5.52 ***
	4.45		4.42	
Supportive DC other	17.56	−4.59 ***	17.65	−4.35 ***
	20.63		20.61	
Delegated DC other	5.81	−3.83 ***	6.12	−1.99 *
	7.20		6.77	
Common DC	17.50	−5.69 ***	16.94	−7.17 ***
	21.12		21.13	
Relationship satisfaction	117.11	−6.84 ***	119.42	−4.78 ***
	133.55		131.53	

* *p* < 0.05; *** *p* < 0.001. Insecure individuals’ scores are in the upper row.

**Table 3 ijerph-19-08026-t003:** Intercorrelations between the variables for women (_A) and men (_P).

		1	2	3	4	5	6	7	8	9	10	11	12	13	14
1	Attachment_A	1													
2	SDC self_A	0.31 **	1												
3	DDC self_A	0.45 **	0.35 **	1											
4	SDC other_A	0.40 **	0.42 **	0.57 **	1										
5	DDC other_A	0.34 **	0.40 **	0.23 **	0.38 **	1									
6	CDC_A	0.47 **	0.52 **	0.78 **	0.36 **	0.70 **	1								
7	Satisfaction_A	0.54 **	0.35 **	0.53 **	0.42 **	0.58 **	0.66 **	1							
8	Attachment_P	**0.40 ****	0.23 *	0.17	0.23 *	0.31 **	0.23 *	0.27 **	1						
9	SDC self_P	0.36 **	**0.24 ****	0.25 **	0.28 **	0.41 **	0.35 **	0.34 **	0.30 **	1					
10	DDC self_P	0.29 **	0.26 **	**0.31 ****	0.14	0.33 **	0.34 **	0.32 **	0.46 **	0.52 **	1				
11	SDC other_P	0.31 **	0.35 **	0.22 *	**0.23 ***	0.35 **	0.31 **	0.29 **	0.38 **	0.59 **	0.68 **	1			
12	DDC other_P	0.04	0.17	0.17	0.00	**0.12**	0.15	0.07	0.18 *	0.39 **	0.43 **	0.46 **	1		
13	CDC_P	0.32 **	0.32 **	0.31 **	0.23 *	0.41 **	**0.41 ****	0.35 **	0.56 **	0.55 **	0.80 **	0.70 **	0.46 **	1	
14	Satisfaction_P	0.31 **	0.12	0.36 **	0.36 **	0.42 **	0.29 **	**0.39 ****	0.41 **	0.33 **	0.41 **	0.42 **	0.14	0.42 **	1

Correlations between spouses are shown in bold diagonal font; *n* = 114 dyads; * *p* < 0.05; ** *p* < 0.01. DC—dyadic coping, SDC—supportive DC, DDC—delegated DC, CDC—common DC.

**Table 4 ijerph-19-08026-t004:** Effects of self-perceived supportive DC mediation.

Effect	Estimate	95% CI	Standardized	Percent Total
Actor				
Total	1.84	1.16 to 2.66	0.30 ***	
Direct	1.16	0.48 to 1.93	0.19 ***	63
Total indirect	0.68	0.31 to 1.09	0.11 ***	37
Actor–actor indirect	0.57	0.29 to 0.95	0.09 ***	31
Partner–partner indirect	0.11	−0.07 to 0.32	0.02	6
Partner				
Total	1.06	0.43 to 1.83	0.17 **	
Direct	0.38	−0.22 to 1.12	0.06	36
Total indirect	0.68	0.29 to 1.05	0.18 ***	64
Actor–partner indirect	0.11	−0.07 to 0.30	0.02	10
Partner–actor indirect	0.57	0.26 to 0.90	0.09 ***	54

** *p* < 0.01; *** *p* < 0.001.

**Table 5 ijerph-19-08026-t005:** Effects of other-perceived supportive DC mediation.

Effect	Estimate	95% CI	Standardized	Percent Total
Actor				
Total	12.80	9.38 to 16.57	0.43 ***	
Direct	8.79	5.76 to 12.24	0.29 ***	69
Total indirect	4.01	2.41 to 6.12	0.13 ***	31
Actor–actor indirect	3.44	1.74 to 5.46	0.11 ***	27
Partner–partner indirect	0.57	0.04 to 1.72	0.02 *	5
Partner				
Total	3.68	0.30 to 7.34	0.12 *	
Direct	0.79	−2.78 to 4.55	0.03	22
Total indirect	2.89	1.06 to 5.13	0.23 **	78
Actor–partner indirect	1.11	0.15 to 2.28	0.04 *	30
Partner–actor indirect	1.78	0.52 to 3.57	0.06 ***	48

* *p* < 0.05; ** *p* < 0.01; *** *p* < 0.001.

**Table 6 ijerph-19-08026-t006:** Effects of self-perceived delegated DC mediation.

Effect	Estimate	95% CI	Standardized	Percent Total
Actor				
Total	12.80	9.26 to 16.17	0.43 ***	
Direct	9.02	5.54 to 13.03	0.30 ***	70
Total indirect	3.79	2.10 to 5.75	0.13 ***	30
Actor–actor indirect	3.48	1.88 to 5.35	0.11 ***	27
Partner–partner indirect	0.31	−0.35 to 0.91	0.01	2
Partner				
Total	3.68	0.61 to 7.28	0.12 *	
Direct	0.97	2.43 to 4.79	0.03	26
Total indirect	2.72	0.89 to 4.39	0.23 ***	74
Actor–partner indirect	2.24	0.67 to 3.90	0.07 **	61
Partner–actor indirect	0.48	−0.52 to 1.55	0.02	13

* *p* < 0.05; ** *p* < 0.01; *** *p* < 0.001.

**Table 7 ijerph-19-08026-t007:** Effects of other-perceived delegated DC mediation.

Effect	Estimate	95% CI	Standardized	Percent Total
Actor				
Total	12.80	9.12 to 16.38	0.43 ***	
Direct	11.30	7.60 to 15.07	0.38 ***	88
Total indirect	1.51	0.39 to 2.80	0.05 **	12
Actor–actor indirect	1.32	0.28 to 2.88	0.04 *	10
Partner–partner indirect	1.18	−0.52 to 0.99	0.006	1
Partner				
Total	3.68	0.34 to 7.31	0.12 *	
Direct	2.39	−0.86 to 6.06	0.08	65
Total indirect	1.29	0.14 to 2.32	0.09 *	35
Actor–partner indirect	1.07	0.17 to 2.24	0.04 *	29
Partner–actor indirect	0.22	−0.86 to 1.16	0.008	6

* *p* < 0.05; ** *p* < 0.01; *** *p* < 0.001.

**Table 8 ijerph-19-08026-t008:** Effects of common DC mediation.

Effect	Estimate	95% CI	Standardized	Percent Total
Actor				
Total	12.80	9.19 to 16.37	0.43 ***	
Direct	7.27	3.69 to 11.06	0.24 ***	57
Total indirect	5.54	3.46 to 8.22	0.18 ***	43
Actor–actor indirect	5.36	3.23 to 7.97	0.18 ***	42
Partner–partner indirect	0.18	−0.16 to 0.76	0.006	1
Partner				
Total	3.68	0.64 to 7.26	0.12 *	
Direct	1.72	−2.10 to 5.39	0.06	47
Total indirect	1.96	−0.02 to 4.08	0.36 *	53
Actor–partner indirect	1.02	−0.81 to 2.93	0.03	28
Partner–actor indirect	0.94	0.33 to 2.31	0.03	26

* *p* < 0.05; *** *p* < 0.001.

## Data Availability

The data that support the findings of this study are available from the corresponding author upon reasonable request.

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
