# Peer review of "Supportive, Delegated, and Common Dyadic Coping Mediates the Association between Adult Attachment Representation and Relationship Satisfaction: A Dyadic Approach"

_ijerph, 2022, doi:10.3390/ijerph19138026_

Round 1
Reviewer 1 Report
The aims of this study was to examine intrapersonal (actor) and interpersonal (partner) associations between secure early parental attachment and marital relationship satisfaction, and to test the mediating role of supportive, delegated and common dyadic coping on the aforementioned association.
INTRODUCTION
The last paragraph (before the section 1.1.1, rows ) must be re-write. The rational of the sentence "Three main attachment categories were identified: secure, insecure ambivalent/anxious, and insecure avoidant [40]" is not clear. You have used an instrument measuring 4 attachment styles. Also, the discussion about the role of attachment figures does not justify the use of early parental attachment instead of romantic or adult attachment (with the partner) in this research. Why is important to study for adults their early parental attachment in relation with their present marital satisfaction? Please provide more consistent arguments.
In the section 1.2. Aim(s) of the study (you present 3 different aims), after the first paragraph of this section, you may underline the originality and the contribution of the present study to the literature. How does your study expand this line of research? In what direction? (e.g, using a community sample versus students samples, using the dimensions of positive DC instead of general DC or total score of positive DC etc.) Also, you may move the second paragraph to the Measure section.
The second and third goals are not clear. Delete the third goal. There are not different analysis for this two goals. In addition, the introduction of a significant mediator leads to the decrease or of the relationship between VI and VD, and not an increase of their association, thus explaining better the link between VI and VD.
You have to present more clearly your goals/aims and hypothesis. For example, in the first hypothesis you mention the secure attachment. What type of attachment was analysed in the second hypothesis? Normally, the secure one, based on previous analysis. Why are you taking then before the second hypothesis about insecure attachment? Did you study the parental insecure attachment too, as predictor of marital satisfaction? Why? I did not find it in results sections.
MATERIALS AND METHOD
Please present more information about construct validity of Relationship satisfaction used measure.
RESULTS
In preliminary analysis, please introduce the table of correlations among all variables of the study, including attachment.
You have mentioned that "the AAI four categories were converted to a dichotomous secure insecure variable." But, in the section 3.1." Attachment as a predictor of relationship satisfaction", you have presented only the secure early parental attachment. In the Figure 2. is missing the word secure attachment.
in the section "3.1.1. Dyadic coping as mediator..." the fact of measuring and reporting data for the secure attachment is totally missing. This is confusing!
DISCUSSION
Please modify and introduce in discussion the above-mentioned suggestions.
Author Response
Dear Reviewer,
Please see the attachment.
Best regards,

Reviewer 2 Report
I have reviewed this manuscript which focuses on the role of attachment and dyadic coping in shaping the relationship satisfaction.
The research is made on a consistent basis and by scientific means. One of the strongest points of the article is that the dyadic approach, implemented in the study, considers the couple as the unit of analysis. It overcomes the analytical strategies of inter and intrapersonal approaches.
As a transactional process, Dyadic coping is well established, so in the positive and negative dimension.
The goals are clearly specified, as are the analysis tools for the variations that are described.
The article is thoroughly researched and rendered with precision. It is extremely well-structured and presents valuable information about the matter. The hypothesis and methodology are well established and conducted. The figures are concise and easy to follow. They nicely illustrate the variables of the text and the process of analysis strategies.
Although it seems slightly local, the authors contemplate the possibility that the conclusions are more universal. Study limitations are well specified.
Regarding Materials and Methods, the participants, procedures, measures (KDM-2, DCI, AAI and Demographic Questionnaire) have been well chosen and described.
In the results, the tables are very accurate and explain the results clearly. The confirmed and partially confirmed hypotheses are supported by the results obtained in a very neat way.
Discussion is interesting, among other things because of the disparate results between men and women. This provides a satisfactory explanation based on previous studies.
I applaud the coherence of the presentation of the limitations in the conclusions section. The authors honestly acknowledge that satisfaction with the relationship is a multidimensional concept that might be affected by multiple factors, so the predictive power of each single factor is necessarily limited.
The study supposes a contribution in the real world, in the clinical application in couple therapy and in the relationship with childhood experiences for the solidity and stability of the relationship when facing a problem or uncertainty or stressful situations.
Although it is not very original, it is a very rich article, in which the problem is very well delimited and the procedural work is explained with useful results.
The bibliography is very complete and adjusted to the subject of study.
I recommend reviewing the typos in lines 97, 143, 487, 506, 528
Overall assessment. Accept after minor adjustments.
Author Response
Dear Reviewer,
We appreciate your positive feedback regarding our work. Your rewarding words are very encouraging and pushing us to more advanced research in the future.
Brest regards,